# Data-Driven Stability Assessment of Multilayer Long Short-Term Memory Networks

Davide Grande [1,*,†], Catherine A. Harris [2,†], Giles Thomas [1,†] and Enrico Anderlini [1,†]

1 Department of Mechanical Engineering, University College London, London WC1E 7JE, UK; giles.thomas@ucl.ac.uk (G.T.); e.anderlini@ucl.ac.uk (E.A.)
2 National Oceanography Centre, Liverpool L3 5DA, UK; catherine.harris@noc.ac.uk
* Correspondence: davide.grande.19@ucl.ac.uk
† These authors contributed equally to this work.

**Abstract:** Recurrent Neural Networks (RNNs) are increasingly being used for model identification, forecasting and control. When identifying physical models with unknown mathematical knowledge of the system, Nonlinear AutoRegressive models with eXogenous inputs (NARX) or Nonlinear AutoRegressive Moving-Average models with eXogenous inputs (NARMAX) methods are typically used. In the context of data-driven control, machine learning algorithms are proven to have comparable performances to advanced control techniques, but lack the properties of the traditional stability theory. This paper illustrates a method to prove a posteriori the stability of a generic neural network, showing its application to the state-of-the-art RNN architecture. The presented method relies on identifying the poles associated with the network designed starting from the input/output data. Providing a framework to guarantee the stability of any neural network architecture combined with the generalisability properties and applicability to different fields can significantly broaden their use in dynamic systems modelling and control.

**Keywords:** multi-layer neural network; recurrent neural networks; system identification; stability analysis

## 1. Introduction

Neural networks are becoming increasingly popular in the fields of dynamic modelling, time series forecasting and control. Deep Neural Networks (DNNs) are employed in control applications when the traditional model based approach lacks design efficiency or is deemed unfeasible. This could be due to the model of the system being very complex, due to a time-varying environment or when the control solution is too cumbersome to compute due to a large action space [1]. When big datasets are available, machine learning algorithms show comparable performance to advanced control techniques such as the combination of real-time optimization and model predictive control [2]. Despite their advantages, data-driven control systems cannot rely on the traditional stability theory used for model based approaches [2]. The lack of stability proofs jeopardises the use of the DNN in safety-critical system identification and control applications in which stability assessment and quantification are of paramount importance.

Among different types of neural networks, Recurrent Neural Networks (RNNs) show promising results compared to more classical DNN architectures due to their ability to remember system features and dynamics correlated with past events [3–5]. Additionally, RNNs have been successfully applied to healthcare and diagnosis, predicting or classifying pathologies [6,7] or risk events based on biosignals [8,9]. The original RNN architectures suffered from the vanishing descending gradient issue, which was successfully addressed in 1997 with the design of the Long Short-Term Memory (LSTM) architecture [10]. Other RNN variants have been proposed, including Echo State Networks (ESNs) [11]. LSTMs, Gated Recurrent Units (GRUs) and ESNs show promising results for dynamic modelling

applications [12], with LSTM being considered the state-of-the-art for tasks involving long-term dynamics learning [13]. In [14], vanilla LSTMs, composed of three gates, a block input, a single cell, an output activation function and peephole connections, were compared with eight different variants. In this comparison, the gates, activation functions and peephole connections were sequentially removed, while the coupling between the input and forget gates (the latter one constituting the GRU network) was added, to identify and isolate the possible limiting features. The authors concluded that the vanilla LSTMs perform reasonably well on benchmark problems such as acoustic modelling, handwriting recognition and polyphonic music modelling. LSTM networks are starting to be applied to linear and nonlinear dynamic system identification, as reported in [15–17]. In these works, the concept of stacking and combining different neural networks layers was exploited. The architecture obtained by stacking several layers of LSTM cells has already been proven to deliver state-of-the-art performances on speech recognition tasks [18,19]. Deep hierarchical architectures, combined with the use of deseasonalised data in a moving window format, typically outperform shallower networks in multiple applications such as paraphrase generation applications [20] and time series forecasting [21].

Despite advances in the fields of deep learning and LSTM, in order to render these techniques suitable for real system identifications and control, some typical stability proprieties are still to be proven. As detailed in [22], RNNs are dynamic systems that need to be analysed for stability, especially when used to design closed-loop control systems. When considering a typical closed-loop scheme, comprising a dynamic system and a control function, a DNN may constitute one or both of the two blocks. Specifically, the application of LSTMs in the system control context is limited by the missing stability properties and equilibria computation [13]. Analytical results regarding the computation of the system equilibria and the stability assessment of a single-layer LSTM were reported in [23]. No work providing analytical formulations or extended algorithms to estimate the system equilibria for deeper architectures appear to have been published.

The method proposed in this paper is designed to have generalisability as a main objective. A multilayer architecture is chosen for this work to illustrate the application in different system identification case scenarios, but any other NN architectures can used in place of the presented one. This paper shows how to assess the stability of a multilayer LSTM network through a black-box analysis, aimed at identifying the system poles. To achieve this, a four step process is detailed, comprising the design of a dataset aimed at exciting different system dynamics, the training and testing of an NN and the analysis of the input/output time series.

To eventually assess the stability of the network, standard concepts from the control theory literature are exploited, such as the identification of the poles and their location in the complex plane (i.e., checking whether they lie in the left-hand side plane for the case of continuous dynamic systems).

This paper shows how this procedure renders the guarantee and benchmark of the stability of LSTMs possible in both a linear and a nonlinear case. Afterwards, this approach is applied to a case of an unstable dynamic system to show the method's ability to detect and quantify system instability in addition to stability.

This work is organised as follows: Section 2 contains the definition of the dynamic models of the benchmark systems, while Section 3 details the system identification method and the software implementation of the framework. Results and discussions are presented in Section 4, alongside the details of the dynamic model and the DNN parameters. Conclusions and future work are reported in Section 5.

## 2. System Dynamic Models

In order to analyse the stability of the neural networks, two standard reference models, well known in the literature, were selected as a benchmark. This allows the poles of the system to be computed analytically. Additionally, it is possible to reproduce the systems in a simulation environment to compare the poles identified from the network to those

of the benchmarks. Poles analysis is typically used in control applications to assess the stability of a system and to infer the dynamic behaviour of the output as a function of different input signals. Analytically, given a generic system input/output couple, the poles represent the roots of the denominator of the transfer function mapping the output dynamic as a function of the input one. A sliding mass-spring-damper system was chosen as an example of a linear system, whereas an oscillating pendulum represents a nonlinear model. In this section, the dynamic systems are described from a theoretical perspective; the state-space descriptions of the models are reported; and an analytical computation of the poles is provided.

### 2.1. Linear Model

The horizontal dynamic equation of a mass-spring-damper system sliding on a surface with no friction is:

$$m\ddot{x} + d\dot{x} + kx = F \tag{1}$$

where $m$ is the value of the sliding mass, $d$ the damping coefficient, $k$ the stiffness coefficient, $F$ the external force applied and $x$ the mass position. The involved quantities and sign convention are reported in Figure 1.

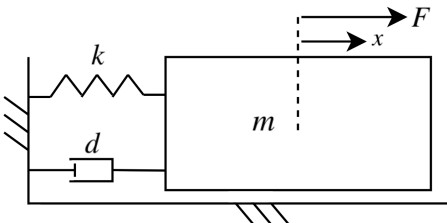

**Figure 1.** Linear system dynamics.

To compute the poles of the system, Equation (1) can be rewritten in state-space form, defining the state-space vector $x = [x_1, x_2]^T$, with $x_1$ the velocity of the mass and $x_2$ its position.

$$\begin{cases} \dot{x}_1 = -\dfrac{d}{m}x_1 - \dfrac{k}{m}x_2 + \dfrac{F}{m} \\ \dot{x}_2 = x_1 \end{cases} \tag{2}$$

The system dynamic matrix of (2), obtained by computing the partial derivatives of $\dot{x}_1$ and $\dot{x}_2$ with respect to $x_1$ and $x_2$, is therefore:

$$A = \begin{bmatrix} -\dfrac{d}{m} & -\dfrac{k}{m} \\ -1 & 0 \end{bmatrix} \tag{3}$$

It is important to notice that, as expected, the equilibria of the system (2) do not depend on the choice of the state space variables, as the system is linear. In order to obtain the analytical formulation of the system poles, it is possible to solve $det(\lambda I - A) = 0$. The resulting poles, $\lambda_1$ and $\lambda_2$, are:

$$\lambda_{1,2} = -\frac{d}{2m} \pm \sqrt{\frac{d^2}{4m^2} - \frac{4}{m}} \tag{4}$$

### 2.2. Nonlinear Model

The same procedure is applied to the benchmark case of a single DOF nonlinear system, i.e., a pendulum rotating around a fixed point with a planar hinge. For the equation describing the rotational dynamics of the system where the total mass of the swinging rod with the mass attached at its extremity is lumped at the end of the former one,

the damping torque depends on the friction in the hinge, and $Mg$ represents the gravity force applied to the mass:

$$J_z\ddot{\theta} + d\dot{\theta} + MgL\sin\theta = T \tag{5}$$

where $z$ denotes the pendulum rotating axis, exiting from the swinging plane, $M$ the value of the mass of the swinging system, $J_z$ its moment of inertia around the $z$-axis, $L$ the length of the rod, $g$ the gravitational constant, $d$ the damping coefficients, $T$ the torque applied to the rod and $\theta$ the pendulum angle, with $\theta = 0$ corresponding to the stable equilibrium of the system. Additionally, it is worthwhile recalling that, based on the assumptions mentioned above, the moment of inertia is computed as $J_z = ML^2$. The parameters and sign convention are reported in Figure 2.

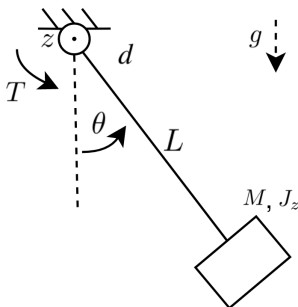

**Figure 2.** Nonlinear system dynamics.

Similarly to the linear case, the state-space form of the pendulum system is described by System (6), with $x_1$ and $x_2$ being the angular position and velocity of the mass, respectively.

$$\begin{cases} \dot{x}_1 = -\dfrac{d}{J_z}x_1 - \dfrac{MgL\sin x_2}{J_z} + \dfrac{T}{J_z} \\ \dot{x}_2 = x_1 \end{cases} \tag{6}$$

The Jacobian evaluated in $x = [0,0]^T$ is reported in Equation (7).

$$A = \begin{bmatrix} -\dfrac{d}{J_z} & -\dfrac{MgL\cos x_2}{J_z} \\ 1 & 0 \end{bmatrix} \tag{7}$$

The system (6) has two equilibria corresponding to the rod position parallel to the gravity force vector. Analytically, the equilibria correspond to the tuple $(x_1^\alpha, x_2^\alpha) = (0\ (\text{rad}), 0\ (\text{rad/s}))$ and $(x_1^\beta, x_2^\beta) = (\pi\ (\text{rad}), 0\ (\text{rad/s}))$ and are denoted by $\alpha$ and $\beta$, respectively. Once more, it is possible to compute the eigenvalues of the system by imposing $det(\lambda I - A) = 0$, with $A$ evaluated in each equilibrium point. The poles' locations associated with $\alpha$ and $\beta$, respectively, are reported hereby.

$$\lambda_{1,2} = -\frac{d}{2J_z} \pm \sqrt{\frac{d^2}{4J_z^2} - \frac{MgL}{J_z}} \tag{8}$$

$$\lambda_{3,4} = -\frac{d}{2J_z} \pm \sqrt{\frac{d^2}{4J_z^2} + \frac{MgL}{J_z}} \tag{9}$$

In absence of the damping term $d$ and recalling that $M$, $g$, $L$ and $J_z$ are always positive, $\lambda_1$ and $\lambda_2$ are complex conjugate, while $\lambda_3$ and $\lambda_4$ are real. It can be inferred that in the undamped case, the eigenvalues associated with $\alpha$ are on the boundary of stability (being purely complex conjugate with $\mathbb{R}e = 0$), while the ones corresponding to $\beta$ are both real, one with $\mathbb{R}e > 0$ and the other with $\mathbb{R}e < 0$. $\lambda_1$ and $\lambda_2$ lead to an oscillating response around the equilibrium point, while $\lambda_3$ and $\lambda_4$ lead to an unstable response. On the other

hand, when $d \neq 0$, the location of the poles and their stability assessment depend on the specific values of the system parameters. In Section 4, two cases are analysed in detail, and the associated poles' values are reported for a specific choice of the parameters.

## 3. Method and Implementation

In this work, a system identification architecture capable of learning and predicting the dynamics of a system is proposed to assess its stability. The designed architecture consists of sequential steps comprising the data generation, the training of a neural network and the computation of the associated poles. The system was designed with the main objective to be as general as possible, in order to allow its application in different fields and with any neural network architectures. In Section 3.1, the methodology of the framework is illustrated, while in Section 3.2, the framework is illustrated in its constituting modules and the software implementation is detailed.

### 3.1. Method

The proposed method consists of the application of a randomly generated dataset to a dynamic system and in the collection of the corresponding output. The designed method treats the block resembling the model dynamics as a black-box, and as such, it is not tied to a specific NN architecture type.

In Figure 3, the architecture used for the identification procedure is depicted. The stability assessment algorithm is designed as a block receiving the input/output datasets of the NN and evaluating the corresponding stability. The same block is applied to the system representing the ground truth dynamics, as it will be used for comparison in Section 4.

The proposed method can be used both offline and online with respect to the training and testing phase of an NN. In this work, the procedure is used offline, since the identification of the location of the poles associated with the network and the assessment of the method's performances were the main aims. The same identification block can be applied online to any other NN (even the ones with the hyperparameters evolving over time) to flag if any unstable dynamics are present or if the boundary of stability is being approached during the training phase.

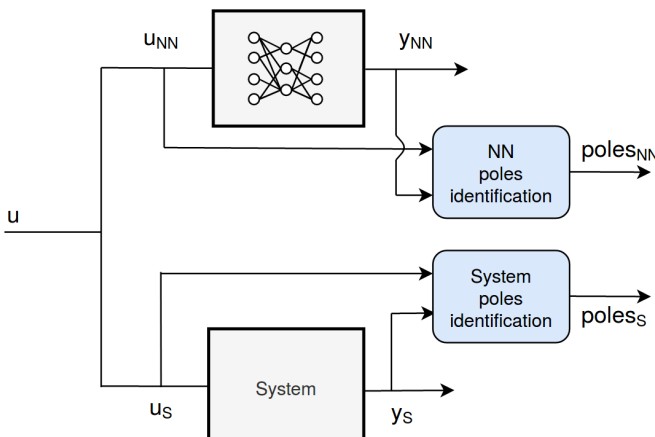

**Figure 3.** System identification method. NN: Neural Network, S: System.

### 3.2. Software Implementation

The framework designed for this implementation of the described methodology was built in MATLAB, Simulink and Python. Simulink was used to simulate the systems and saving the datasets, while Python was used to design, train and test the LSTMs. MATLAB served as the main source of the interface to launch the dynamic models, to distribute the datasets, to execute the callbacks to the LSTM scripts, to post-process the results and to identify the system poles. This mixed architecture allows exploiting the optimal functionalities of both systems. A will be further detailed in this section, Python provides well-established

open-source machine learning libraries, whereas the poles' identification procedure is rendered possible by functions belonging to the MATLAB System Identification Toolbox (https://uk.mathworks.com/help/ident/). The framework follows four logical steps:

1. A sequence of diverse inputs is generated and fed to the model simulated in a MATLAB/Simulink framework. The output signal is saved and together with the input data constitutes the dataset;
2. The LSTM network is trained with the input/output dataset previously generated;
3. A new dataset is generated and fed both to Simulink and to the LSTM;
4. The datasets are analysed with built-in MATLAB functions, and the poles are identified.

The framework architecture is illustrated in Figure 4.

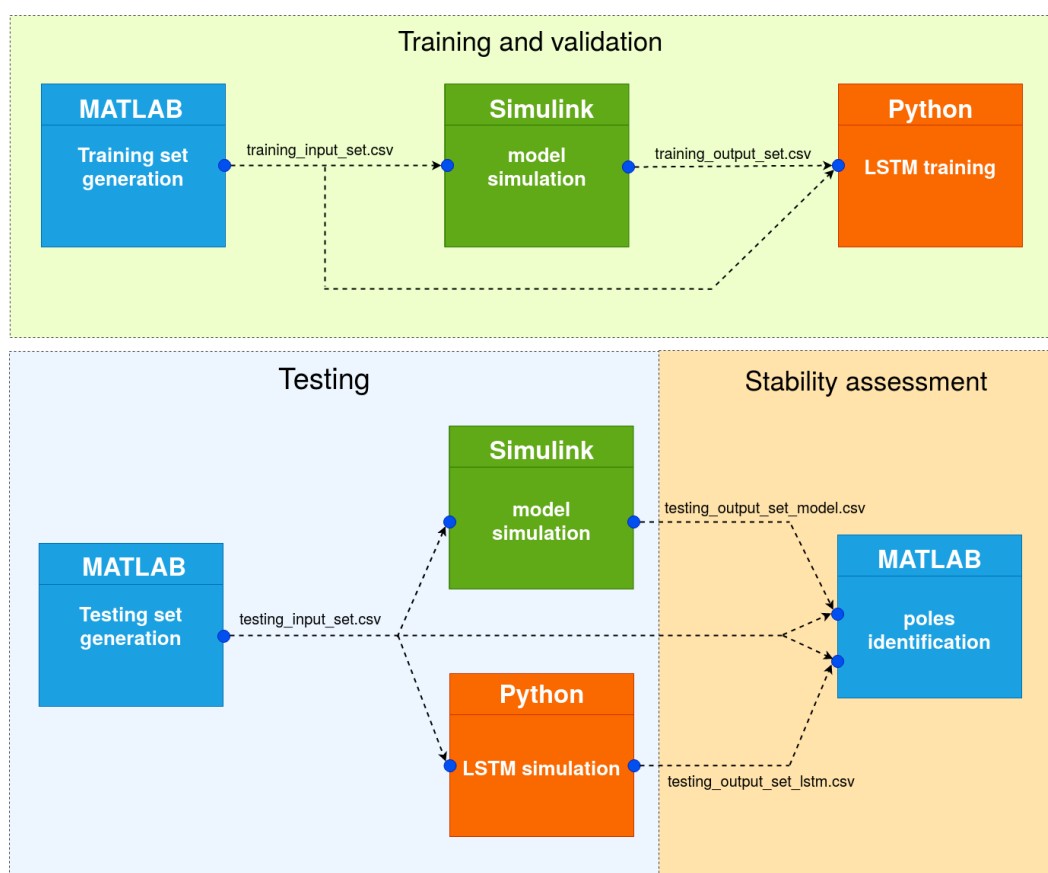

**Figure 4.** Framework architecture—dashed lines: datasets flow, blue dots: modules interfaces.

3.2.1. Training Dataset Generation

The input dataset is generated by recursively selecting a step, a ramp or a sinusoidal input for a fixed amount of time. The input dataset is a single time series comprising the series of runs, where a run corresponds to the selection of one input. For each run, the input is randomly selected as one of the three available signals. The magnitude of the signals, the slopes of the ramps and the frequencies of the sinusoidal inputs are randomly selected within pre-tuned boundaries.

Having different inputs proves to be a key factor in exciting as many system dynamics as possible and in identifying the system poles. Additionally, having a large dataset is another fundamental aspects to ensure the training of the LSTM is effective.

The size of the datasets and practical considerations on how to select the inputs are reported in Section 4.6.

The models are simulated with a fixed step size, required by the following identification procedure.

### 3.2.2. Neural Network Definition and Training

As mentioned in Section 1, stacking neural network layers can improve the identification performance. For this application, the chosen architecture comprises two stacked LSTM cells and a top multilayer perceptron. This hierarchical structure proved to increase the estimation quality with respect to using both simple feedforward and vanilla LSTM architectures in non-linear system identification applications [16]. Improving the depth of the networks is proven to further augment the capability of learning and consequently improving the forecasting capabilities.

Additionally, including a look-back factor proves to significantly improve the performances of the system. This allows the network to predict the next output value based on a pre-defined number of previous points of the input signal.

The DNNs were designed in Python using the Keras (https://keras.io/) library as a front-end, in turn based on the TensorFlow (https://www.tensorflow.org/) library as a back-end.

The details concerning the dimension of the network, the training and the look-back factor are reported in Section 4.1.

### 3.2.3. Testing Dataset Generation

The testing dataset generation follows the same criteria detailed for the training set. Several runs are generated constituted by random inputs with different signal properties. The input time series is provided to both the Simulink model and to the trained neural network to compare the results.

### 3.2.4. Poles' Identification and Stability Assessment

Following the generation of the datasets, the poles are identified using built-in MATLAB functions. *tfest* (https://www.mathworks.com/help/ident/ref/tfest.html) (transfer function estimation) is used as the main identification means. The function provides versatile estimation parameters and does not require any a priori knowledge of the system. *tfest* just requires the input and output time series, the sampling time and the number of zeros and poles of the transfer function to be estimated. This might require a trial-and-error step while identifying systems where a dynamic model, and in turn the order of the system, is not known. The cases analysed in this paper are all second order systems.

Additionally, in the case of underdamped system, when complex-valued poles are expected to be estimated, the *procest* (https://www.mathworks.com/help/ident/ref/procest.html) (process estimation) function can be used. This improves the accuracy of the location of the imaginary part of the poles, but the application is limited to systems of a maximum third order.

Eventually, splitting the dataset and identifying the poles associated with every run prove to be an effective means to increase the estimation accuracy. The dataset is divided into input/output sequences associated with each input; the poles are identified and, then, the results averaged over the entire dataset.

## 4. Results and Discussion

Hereby, the parameters of the dynamic models and the DNN are reported, alongside the results of the identification procedure. In detail:

1. The tuning of the parameters of the network is explained in Section 4.1;
2. The physical parameters of the models are introduced, and theoretical poles obtained with that parameter set are reported in Section 4.2;
3. The statistics of the identified poles are given and compared to the theoretical ones in Section 4.3;
4. Additional analyses are reported to integrate the presented results in Sections 4.4 and 4.5;
5. Lessons learned and additional design considerations are reported in Section 4.6.

### 4.1. Tuning of the Network Parameters

When tuning the DNN architecture described in Section 3.2.2, the tuned parameters are:

- the dimension of LSTM cells;
- the number of training epochs;
- the look-back factor.

Despite advancement in the use and understanding of LSTM networks, no analytical procedure is yet available to tune the dimension and training parameters. They are tuned in trial-and-error steps, fixing two parameters and modifying one at a time. The goal is to minimise the selected loss function, the Mean Squared Error (MSE). The MSE, computed at each epoch, is defined as:

$$MSE = \frac{1}{n} \sum_{i=1}^{n} (Y_i - \hat{Y}_i)^2 \tag{10}$$

where $n$ is the total number of points provided at each epoch and $(Y_i - \hat{Y}_i)$ represents the difference between the output of the Simulink model and the one of the DNN.

The dimension of each LSTM layer was tested starting from a dimension of two and increasing the size following an exponential of base two distribution. The optimal network size that minimised the MSE was found to be 128 cells.

The number of training epochs that produced the best fit to the reference signal without overfitting was 100. This was tuned starting from 10 epochs and doubling the number of epochs while evaluating the MSE and the computational time, which was chosen not to exceed 5 h of training.

Finally, the look-back factor was tuned. The best performance was obtained when it was 100, meaning that generating the dataset at 10 Hz, the DNN predicted the next output based on the last 10 s of the input signal.

To carry out the training and validation of the network, eighty percent of the dataset was used for training and the remaining 20% for validation. In order to avoid the saturation of the input gate, the input time series was standardised based on the mean value and the standard deviation of the training set. This avoids any leakage of information coming from the validation set. An example of the model fitting is reported in Figure 5.

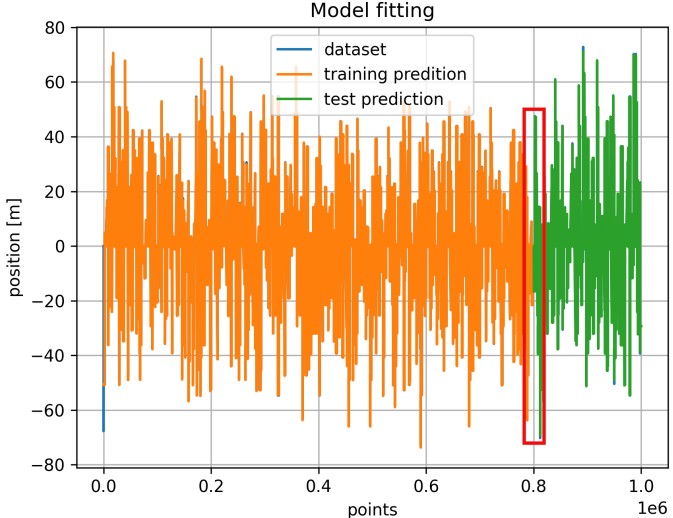

**Figure 5.** Simulink output dataset (blue) vs. DNN training prediction set (orange) and DNN validation prediction set (green).

It should be noted that the output time series predicted (corresponding to the training and the validation sets, reported in orange and green, respectively) almost entirely superimpose the Simulink generated set (drawn in blue).

A zoomed plot, corresponding to the red box highlighted in Figure 5, is reported in Figure 6. This illustrates the section where the two datasets switch: the last input corresponding to the training set is a sinusoidal input, and the first one to constitute the validation set is a ramp with a negative slope. The effect of the look-back factor can also be observed: the first 100 points of the validation set are not predicted since they are needed to estimate the first output value.

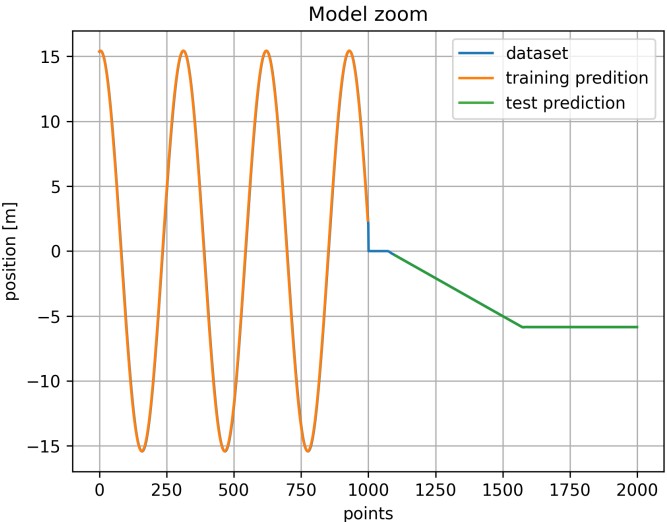

**Figure 6.** Network output vs. dataset zoom.

In Figure 7, the loss function is reported as an example for the training of the nonlinear model described by Equation (5), with the optimal network parameters mentioned above.

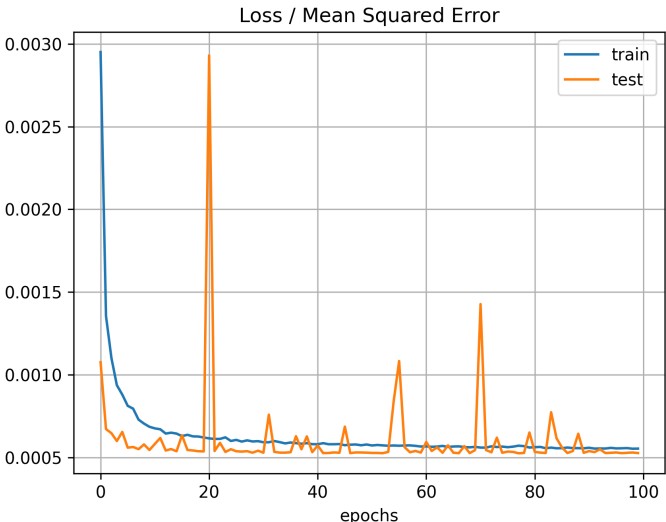

**Figure 7.** Loss function.

### 4.2. Tuning of the Model Parameters and Theoretical Poles

Two model parameter sets were selected in order to prove how the poles' estimation procedure works in the presence of different system dynamic behaviour: for both the models defined by Equation (1) and Equation (5), two DNN were trained, corresponding to the underdamped and the overdamped responses. To obtain the desired dynamics, all the parameters of model (1) were fixed to be equal to one, apart from the damping coefficient $d$, and based on the poles described in Equation (4), $d$ was chosen to obtain either real or complex conjugate poles. The same process was repeated for the nonlinear model (5). The system response was validated with the use of *pplane* (https://www.mathworks.com/matlabcentral/fileexchange/61636-pplane). Figures 8 and 9 show the state-space

trajectories of the linear model in the underdamped ($d = 1$ kg/s) and in the overdamped ($d = 10$ kg/s) case, respectively.

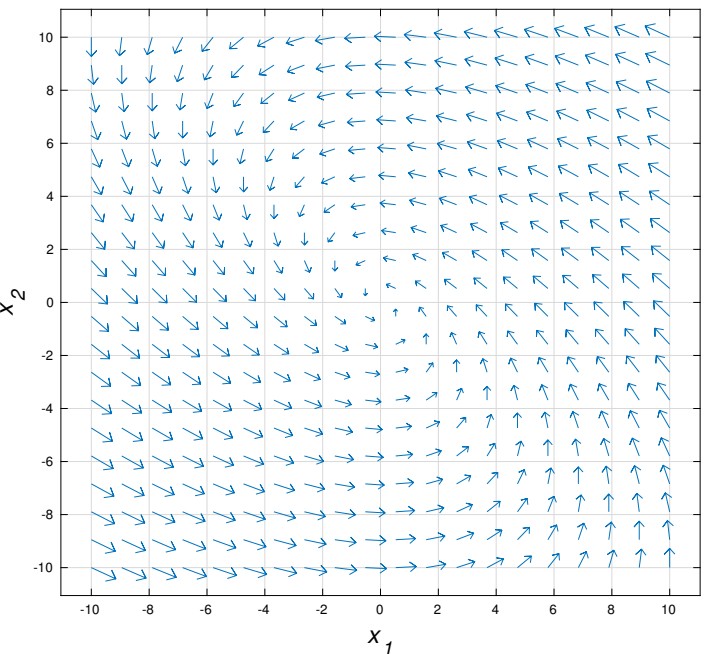

**Figure 8.** Linear model—underdamped.

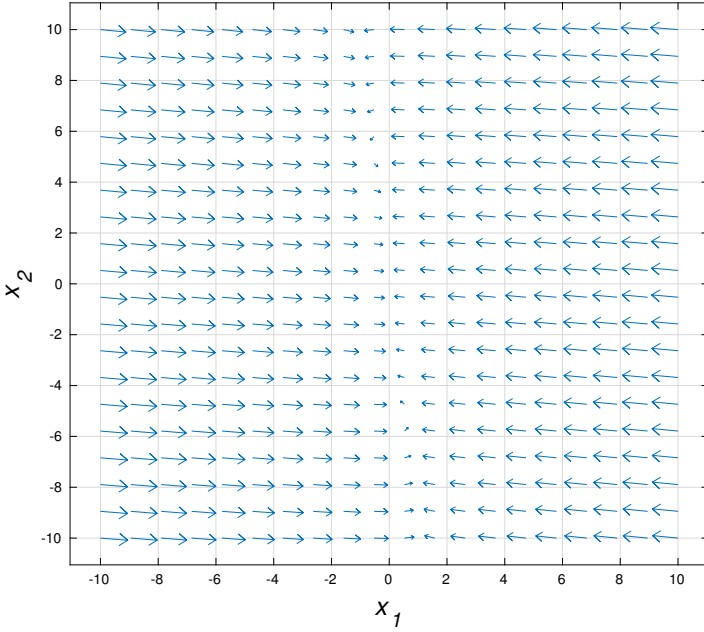

**Figure 9.** Linear model—overdamped.

The same procedure was applied for the nonlinear case, where the parameters were set as follows: $M = 1$ kg, $L = 1$ m, $g = 9.81$ m/s$^2$, $d = 2$ kg/s for the underdamped case and $d = 7$ kg/s to obtain an overdamped dynamics.

### 4.3. Poles' Statistics in Different Models

In this section, the theoretical poles obtained from the two considered models and the parameters are reported and the statistics generated.

Table 1 summarises the poles' locations for the two models for the parameter sets described.

**Table 1.** Theoretical poles' locations.

| Model | Damping | Pole 1 (rad/s) | Pole 2 (rad/s) |
|---|---|---|---|
| linear | underdamped | $-0.50 + 0.87j$ | $-0.50 - 0.87j$ |
| linear | overdamped | $-0.10$ | $-9.90$ |
| nonlinear | underdamped | $-1.00 + 2.97j$ | $-1.00 - 2.97j$ |
| nonlinear | overdamped | $-1.94$ | $-5.06$ |

The statistics are reported in Tables 2 and 3, in the form of the relative error between the theoretical poles and the ones estimated from the generated datasets for the overdamped and underdamped dynamics, respectively.

**Table 2.** Poles' location statistics: overdamped models.

| Dynamics | Model | Slow Pole | Fast Pole |
|---|---|---|---|
| linear | Simulink | 50.01% | 15.57% |
| overdamped | LSTM | 34.77% | 92.08% |
| nonlinear | Simulink | 7.67% | 21.99% |
| overdamped | LSTM | 43.29% | 47.89% |

**Table 3.** Poles' location statistics: underdamped models.

| Dynamics | Model | $\mathbb{R}e$ | $\mathbb{I}m$ |
|---|---|---|---|
| linear | Simulink | 0.17% | 1.64% |
| underdamped | LSTM | 74.2% | 26.5% |
| nonlinear | Simulink | 4.38% | 4.38% |
| underdamped | LSTM | 50.86% | 3.93% |

*4.4. Unstable Model Identification*

The procedure has so far been applied to stable models. In this section, the framework is tested in the presence of data generated by an unstable process. The chosen model is a second order system with a pole lying in the right-hand side of the complex plane and described by the following transfer function:

$$\frac{1}{(s+10)(s-0.01)} \tag{11}$$

The unstable pole is defined as $\lambda = 0.01$, while the stable one is set as $\lambda = -10$. Additionally, *tfest* assumes by default that the time series used for the identification procedure are generated by stable processes. This implies that, without modifying its settings, the poles are identified in the interval $(-\infty, 0]$. This issue can be overcome by modifying the stability threshold, normally set as $s = 0$ for continuous-time systems and $z = 1$ for discrete-time ones.

In this example, the corresponding parameter to be modified is set as *tfestOptions. Advanced.StabilityThreshold.s* $= 10^3$. Once more, the model is simulated and the corresponding DNN trained and tested. The distribution of the identified poles is reported in Figure 10. It can be seen that out of 4000 runs, 2693 times (0.67%), the real part of the unstable poles was correctly identified in the right-hand side of the complex plane. The framework was capable of recognizing that an unstable dynamics arises in the model generating the data and flagging the behaviour accordingly. As described in Section 1, this framework can be used to detect instabilities linked to any NN architecture, even the ones whose weights evolve over time (e.g., the ones associated with Reinforcement Learning

(RL) algorithms). The process generating the data in this example, i.e., Equation (11), is time-invariant, but the same procedure applied to time-variant systems would be able to detect an arising instability online. In fact, any single pole identified as unstable is enough to trigger the detection algorithm and the corresponding notification that the process generation is no longer stable. To conclude, the poles wrongly detected as stable are the ones associated with runs where ramp signals are provided as inputs: the ramps are in fact the signals generating the slowest dynamic responses and not always triggering an unstable behaviour fast enough to be identified. Additional considerations on the issue are detailed in Section 4.6.

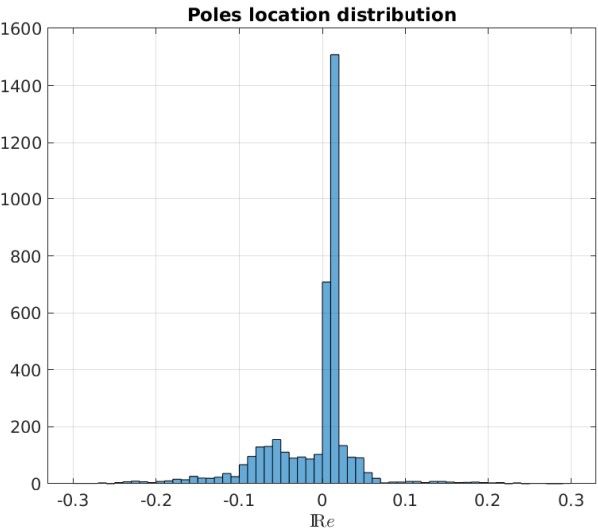

**Figure 10.** Unstable model—poles' locations.

### 4.5. Introducing Noise

After having shown the results for ideal cases in which the datasets are generated by processes with no noise (or equivalently in which the noise has been filtered out), in this section, one of the benchmark cases is analysed, introducing disturbances in the data generation process.

The chosen model is the nonlinear system defined by Equation (5). The disturbance is introduced in the form of a noisy output variable, resembling a real case in which the reading of the swinging mass angle $\theta$ is provided by a non-ideal sensor. The signal-to-noise ratio was chosen as 25 dB, ensuring a noise power greater than the signal power. A comparison between the testing datasets is reported in Figure 11, with the output of the Simulink model and the DNN drawn in blue and red, respectively.

It is possible to notice that the DNN is able to identify the model dynamics, and it additionally shows intrinsic filtering capabilities, both in the steady-state and in the transient sections of the responses.

The poles' locations were then evaluated. The slow poles of both Simulink and the DNN were estimated with a maximum relative error bounded below 50%, showing comparable performances with respect to the ideal case. The fast pole is on the contrary identified showing a loss of accuracy performance (several orders of magnitude greater, but still identifying that the system is stable).

It is important to bear in mind that in a real-case scenario involving a noisy signal, before the training step, an intermediate filtering stage would be performed.

The analysis reported in this section shows how the designed framework can deal with system noise: if the aim of the system identification is only to assess the stability of a process, the system does not require additional steps, whereas if the purpose is to accurately locate the position of the poles, a filtering stage is suggested. If the noise properties are constant (e.g., its variance), it is possible to change the numerical search method, for instance by

setting *tfestOptions.SearchMethod* equal to *fmincon lm*, or *gn*, with the advantage of avoiding the filtering stage.

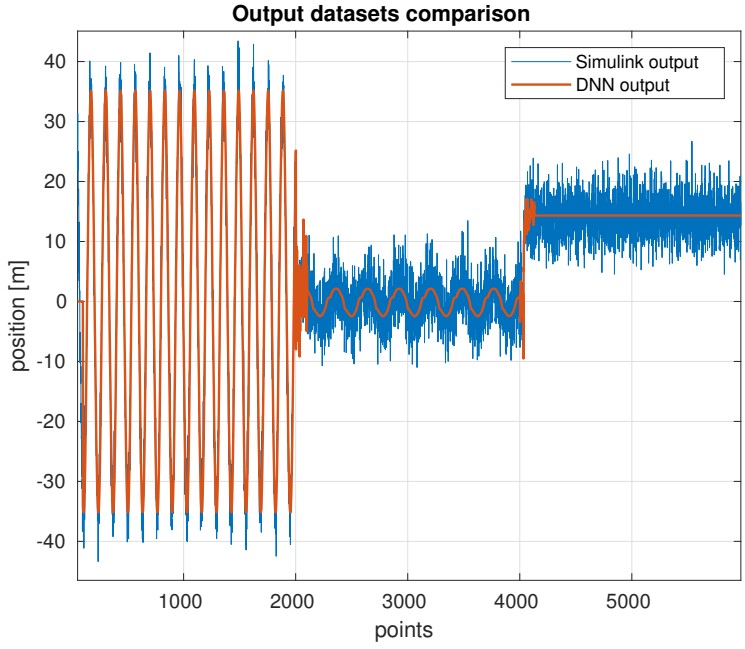

**Figure 11.** Nonlinear model (overdamped case)—system with noise.

### 4.6. Additional Design Considerations

In this section, practical considerations regarding the generation and the scaling of the dataset, as well as the choice of the input signals are reported.

1.  Ensuring that the training dataset is composed by different signals in terms of magnitude and type is a key factor to obtain an NN capable of generalising the output. Additionally, other experimental results proved to be key to improve the ability of the network to learn different dynamic behaviour with corresponding improvements in the associated poles. During the design of the datasets, two main factors emerged, which are capable of jeopardising the accuracy of the identified poles:

    *   the presence of ramps with low slopes;
    *   the presence of sinusoidal inputs with long periods.

    More generally, the presence of inputs exciting only slow dynamics is believed to cause issues while using *tfest*. To avoid the issues associated with the sinusoidal inputs, it is possible to set a lower boundary to the period of those signals. By defining the period of the chosen sinusoidal inputs $T_{sin}$ and the period of the simulation run $T_{run}$, it is possible to impose: $T_{sin} \geqslant T_{run}$.
    Regarding the slope of the ramp ($m_r$), no analytical boundary was identified, and a trial-and-error approach led to finding different values for each model simulated (for reference: $m_r \geqslant 0.1$ (N) and $m_r \geqslant 0.5$ (Nm) for the linear and nonlinear models, respectively).

2.  It is also important to recall that the dataset size affects the training capabilities of the DNN. In this application, the dataset, formed by 500 runs (corresponding to 500 different inputs), has $10^6$ points. Previous training tests carried out with a smaller dataset did not show satisfactory identification performances. Ensuring that the training set is large enough is therefore an important aspect of the procedure.

3.  The last factor affecting the performance of the framework is identified as the scaling of the dataset. The input gate of the LSTM cells is sensible relative to the magnitude of the input signal: if the latter is excessive, the gate can saturate and show degrading identification performances. The input time series can be bounded between $[0, 1]$ or $\pm 3\sigma$ (with $\sigma$ denoting the standard deviation) by means of a normalisation or

a standardisation, respectively. For this work, the dataset was standardised since the method proved to be more robust with respect to the presence of outstanding outliers. For instance, the presence of a single step signal with a final value one order of magnitude higher than the average value of the rest of the dataset in a normalised dataset could lead the NN to assign a higher weight to some neurons in order to minimise the error for that case. That would in turn cause a degradation of the performance for the signals with a smaller magnitude. On the contrary, the standardisation of the dataset allows detecting when the high value signal is outside the $\pm 3\sigma$ interval of the dataset and consequently assigning to it a relative lower weight during the training step.

## 5. Conclusions

In this paper, a new procedure to compute the poles of a generalised DNN architecture is proposed. The procedure consists of generating a dataset composed of different inputs in order to excite as many dynamics of the network as possible. The poles are then extracted from the time series by analysing their input/output dynamic relationship. The procedure generates accurate results for the estimation of the location of the poles and is shown to be able to assess the stability of the systems in all the analysed cases. Starting from a linear system and having obtained positive results, a nonlinear standard reference model was designed to check whether the accuracy of the poles' location is preserved even in more realistic case scenarios, with consistent outcomes. Following, an unstable system is chosen as source of data generation, and a DNN is trained to learn the dynamics and to assess the stability: the framework proves to detect and flag its instability. Eventually, the performances of the identification procedure are reported in the presence of noisy signals, and the corresponding degradation of the performance is quantified.

Since the procedure is general and not tied to the specific network architecture used in this work (i.e., stacked LSTM layers), the authors are confident that it can be applied to other DNN architectures with consistent results. The contribution of this paper consists of mixing standard concepts and techniques used in system identification and in their application to a newer field, i.e., in the identification of dynamic models with DNN and in the assessment of their stability. Previous approaches proposed in the literature showed the stability properties of specific DNN architectures, whereas this paper presents a framework that can be applied to architectures of arbitrary complexities. Other works rely on grey-box system identification techniques, for which the parametric description of the model is available. In contrast, in this work, a black-box system identification procedure is developed. This renders the described approach applicable to any architectures where the analytical description of the system is too difficult to obtain or has not yet been developed.

The main application of interest is represented by the identification of the poles of the network implementing learning control algorithms, for which the stability remains a fundamental property yet to be proven analytically. RL algorithms are based on DNNs evolving over time, and the criteria defined in this paper to assess the stability of the DNN can be used online in RL applications. The framework designed in this work can assess the stability of any DNN architecture online by iterative computations of the system poles. It is suited to be applied as an online supervisor to control systems and flag whether the system is approaching the boundary of stability by recursively computing the system poles. The limitation of the proposed work, to be proven with future analyses, is in the performance assessment of the real-time identification of the system poles. Additional developments will consist of extending the stability analysis shown in the two single-input single-output systems to the case of multiple-input multiple-output ones. Further works will comprise the application of the procedure to data generated by real dynamic systems for which the stability cannot be easily assessed (e.g., for autonomous underwater gliders) and to feedback systems constituted by one or more DNN.

**Author Contributions:** Conceptualization, D.G., E.A., C.A.H. and G.T.; methodology, D.G., E.A. and C.A.H.; software, D.G. and E.A.; validation, E.A.; formal analysis, D.G., E.A. and C.A.H.; investigation, D.G., E.A. and C.A.H.; resources, D.G., E.A. and C.A.H.; data curation, D.G., E.A.; writing—original draft preparation, D.G.; writing—review and editing, E.A., C.A.H. and G.T.; visualization, D.G.; supervision, E.A., C.A.H. and G.T.; project administration, E.A.; funding acquisition, C.A.H., E.A. and G.T. All authors read and agreed to the published version of the manuscript.

**Funding:** This research received no external funding.

**Institutional Review Board Statement:** Not applicable.

**Informed Consent Statement:** Not applicable.

**Data Availability Statement:** Not applicable.

**Acknowledgments:** D. Grande's studentship was supported by the National Oceanography Centre and by the University College London.

**Conflicts of Interest:** The authors declare no conflict of interest.

## Abbreviations

The following abbreviations are used in this manuscript:

| | |
|---|---|
| RNN | Recurrent Neural Networks |
| NARX | Nonlinear AutoRegressive models with eXogenous inputs |
| NARMAX | Nonlinear AutoRegressive Moving-Average with eXogenous inputs |
| DNN | Deep Neural Networks |
| ESN | Echo State Networks |
| GRU | Gated Recurrent Units |
| MSE | Mean Square Error |
| RL | Reinforcement Learning |

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
