# Peer review of "Data-Driven Stability Assessment of Multilayer Long Short-Term Memory Networks"

_applsci, doi:10.3390/app11041829_

Round 1

Reviewer 1 Report

The study carried out is novel. The focus of the study is original and practical. The methodology used is correctly described in the manuscript and developed enough to allow the reproducibility of the study. I just have a few suggestions:

First, the introduction indicates some of the most popular Recurrent Neural network applications today. I also recommend mentioning its application in the area of healthcare and diagnosis, predicting or classifying pathologies [1,2] or risk events based on biosignals [3,4].

I suggest a further explanation of the pole concept at the beginning of section 2, since it is the key aspect of the study, deepening the relevance of its calculation to achieve greater stability of the system. This will help readers to understand the relevance and scope of the proposed framework from the beginning of reading the manuscript.

Finally, the limitations of this method and the study should be discussed, since it depends on having an analytical model that allows the calculation of the poles. These limitations are implicitly noted in the last sentence of the conclusions, indicating that it is a possible future work, but no ideas are provided that would encourage other researchers to delve into this line of research that this study introduces.

[1] Cheng, Maowei, et al. "Recurrent neural network based classification of ECG signal features for obstruction of sleep apnea detection." 2017 IEEE International Conference on Computational Science and Engineering (CSE) and IEEE International Conference on Embedded and Ubiquitous Computing (EUC). Vol. 2. IEEE, 2017.

[2] Shahid, Farah, Aneela Zameer, and Muhammad Muneeb. "Predictions for COVID-19 with deep learning models of LSTM, GRU and Bi-LSTM." Chaos, Solitons & Fractals 140 (2020): 110212.

[3] Luna-Perejón, Francisco, Manuel Jesús Domínguez-Morales, and Antón Civit-Balcells. "Wearable fall detector using recurrent neural networks." Sensors 19.22 (2019): 4885. Luna-Perejón, Francisco, Manuel Jesús Domínguez-Morales, and Antón Civit-Balcells. "Wearable fall detector using recurrent neural networks." Sensors 19.22 (2019): 4885.

[4] Ma, Liang, et al. "Room-level fall detection based on ultra-wideband (UWB) monostatic radar and convolutional long short-term memory (LSTM)." Sensors 20.4 (2020): 1105.

Reviewer 2 Report

The paper is written by suitable way. The content of the paper is accetable. I have only a few suggestions for authors:

  1. Page 3: you should also consider friction force for the model. The same point is for the next nonlinear model.
  2. Why you have used MATLAB as well as Python for your software ? MATLAB as well as Python are suitable for whole task from simulation to training and test. 
  3. Page 9: There should be written units of M, L.
  4. Conclusion: Please, try to describe the novelty of your paper with higher emphasis. Be clear, what is the difference of your approach. 
